# Genetic Attenuation of Paraoxonase 1 Activity Induces Proatherogenic Changes in Plasma Proteomes of Mice and Humans

**DOI:** 10.3390/antiox9121198

**Published:** 2020-11-28

**Authors:** Marta Sikora, Ewa Bretes, Joanna Perła-Kaján, Izabela Lewandowska, Łukasz Marczak, Hieronim Jakubowski

**Affiliations:** 1European Centre for Bioinformatics and Genomics, Institute of Bioorganic Chemistry, 61-704 Poznań, Poland; mperdziak@yahoo.com (M.S.); izek1988@gmail.com (I.L.); lukasmar@ibch.poznan.pl (Ł.M.); 2Department of Biochemistry and Biotechnology, Poznań University of Life Sciences, 60-632 Poznań, Poland; ewa.bretes@up.poznan.pl (E.B.); asiape1@gmail.com (J.P.-K.); 3International Center for Public Health, Department of Microbiology, Biochemistry and Molecular Genetics, Rutgers University-New Jersey Medical School, Newark, NJ 07103, USA

**Keywords:** *PON1* genotype, plasma proteomes, lipoproteins, atherosclerosis, humans, mice

## Abstract

High-density lipoprotein (HDL), in addition to promoting reverse cholesterol transport, possesses anti-inflammatory, antioxidative, and antithrombotic activities. Paraoxonase 1 (PON1), carried on HDL in the blood, can contribute to these antiatherogenic activities. The *PON1*-*Q192R* polymorphism involves a change from glutamine (Q variant) to arginine (R variant) at position 192 of the PON1 protein and affects its enzymatic activity. The molecular basis of PON1 association with cardiovascular and neurological diseases is not fully understood. To get insight into the function of PON1 in human disease, we examined how genetic attenuation of PON1 levels/activity affect plasma proteomes of mice and humans. Healthy participants (48.9 years old, 50% women) were randomly recruited from the Poznań population. Four-month-old *Pon1*^−/−^ (*n* = 17) and *Pon1*^+/+^ (*n* = 8) mice (50% female) were used in these experiments. Plasma proteomes were analyzed using label-free mass spectrometry. Bioinformatics analysis was carried out using the Ingenuity Pathway Analysis (IPA) resources. *PON1-Q192R* polymorphism and *Pon1*^−/−^ genotype induced similar changes in plasma proteomes of humans and mice, respectively. The top molecular network, identified by IPA, affected by these changes involved proteins participating in lipoprotein metabolism. Other *PON1* genotype-dependent proteomic changes affect different biological networks in humans and mice: “cardiovascular, neurological disease, organismal injury/abnormalities” in *PON1-192QQ* humans and “humoral immune response, inflammatory response, protein synthesis” and “cell-to-cell signaling/interaction, hematological system development/function, immune cell trafficking” in *Pon1*^−/−^ mice. Our findings suggest that PON1 interacts with molecular pathways involved in lipoprotein metabolism, acute/inflammatory response, and complement/blood coagulation that are essential for blood homeostasis. Modulation of those interactions by the *PON1* genotype can account for its association with cardiovascular and neurological diseases.

## 1. Introduction

Inflammation promotes several key events during the development of atherosclerosis. High-density lipoprotein (HDL), in addition to promoting reverse cholesterol transport, has also been reported to possess anti-inflammatory, antioxidative, and antithrombotic activities important for its atheroprotective function [1]. A calcium-dependent hydrolytic enzyme, paraoxonase 1 (PON1), carried on HDL in the blood, has been suggested to contribute to these antiatherogenic activities [2,3]. The *PON1* gene has several polymorphisms, including the Q192R, which involves a change from glutamine (Q variant) to arginine (R variant) at position 192 of the amino acid sequence of the PON1 protein and affects its hydrolytic activity [4]. Historically, the hydrolytic activity of the PON1 enzyme has been measured with non-natural substrates such as the organophosphate paraoxon (for which the PON1 enzyme has been named) and phenyl acetate, which have been considered as surrogates for an unknown endogenous substrate that promotes atherogenesis [5]. One such endogenous substrate of PON1, homocysteine thiolactone [6], which can adversely impact protein structure/function by posttranslational modification [7], has been recently shown to be a predictor of myocardial infarction in coronary artery disease patients [8]. Levels of Hcy-thiolactone are higher in low-activity *PON1-192QQ* than in high-activity *PON1-192RR* individuals [9]. Hcy-thiolactonase and paraoxonase (POase) activities of PON1 are strongly correlated [10,11], suggesting that the nonphysiological POase is a good surrogate for the physiological Hcy-thiolactonase activity.

Accumulating evidence from human studies and animal models suggests that antiatherogenic and cardioprotective effects of HDL are largely attributable to PON1 [12]. For example, induction of hyperlipidemia, either by a high-fat diet or *ApoE* gene deletion, increases atherosclerotic lesion size in *Pon1^−/−^* mice compared with *Pon1^+/+^* animals [13,14]. Overexpression of the human *PON1* transgene in these animals reduces aortic lesion size and decreases epitopes recognized by antibodies specific for oxidized lipid-protein adducts [15]. In humans, individuals with the low activity *PON1-192QQ* genotype have an increased risk of all-cause mortality and of major adverse cardiac events [16]. Individuals with the highest PON1 activity quartile, the majority of which have *PON1-192RR* genotype, had lower incidence of major adverse cardiac events compared with those in the lowest activity quartile. PON1 activity is associated with cardiovascular risk independent of several established risk factors except for HDL cholesterol [17]. This association is attenuated by adjusting for HDL cholesterol, which is expected because PON1 is an important component of HDL.

PON1 has also been implicated in Alzheimer’s disease (AD) [18], which is expected given that AD has a significant vascular component. For example, genomic association studies show that a single-nucleotide polymorphism in the *PON1* gene is a risk factor for AD [19]. Other studies show that low PON1 activity is also linked to the risk of AD and dementia [20,21]. In mice, deletion of the *Pon1^−/−^* gene makes the animals overly sensitive to the neurotoxicity of Hcy-thiolactone [22].

The molecular basis of the involvement of PON1 in CVD and AD is not fully understood. To get insight into the function of PON1 in human disease, in the present work we have used label-free mass spectrometry and the Ingenuity Pathway Analysis to examine how genetic attenuation of PON1 levels/activity affect plasma proteomes of mice and humans.

## 2. Materials and Methods

### 2.1. Participants

The present study included healthy participants randomly selected from the Poznań population (*n* = 100, 48.9 years old, 48.9% women). Participant characteristics for the whole group and for each *PON1-Q192R* polymorphism are described in Appendix A**.** All participants had signed an informed consent. The study was approved by the Bioethics Committee of the Poznań University of Medical Sciences (approval No. 661/16, approved 16 June 2016).

### 2.2. Mice and Diet

*Pon1*^−/−^ mice [13] and wild-type *Pon1*^+/+^ littermates (C57BL/6J) were maintained at the Rutgers University, New Jersey Medical School Animal Facility. Mice of both sexes (*Pon1*^−/−^, *n* = 17; *Pon1*^+/+^, *n* = 8) were fed a normal rodent chow (LabDiet 5010, Purina Mills International, St. Louis, MO, USA). The animal procedures were approved by the Institutional Animal Care and Use Committee (Protocol #16114D0320, approved 29 March 2017).

### 2.3. Blood Collection

Human blood was drawn by venous puncture into EDTA tubes. Mouse blood was collected from the cheek vein into Eppendorf tubes containing 1% (*v/v*) 0.5 M EDTA. After centrifugation (2000× *g*, 15 min, 4 °C), cleared plasma and cell pellets were frozen at −80 °C. Blood was also collected into serum tubes, allowed to clot for 30 min, serum separated, and stored as above.

### 2.4. Genotyping

Genomic DNA was extracted from whole human or mouse blood using the phenol extraction method and stored at −80 °C. The human PON1 locus was genotyped by PCR-RFLP as previously described [10] following the procedure of Humbert et al. [23]. The *PON1-192* alleles were amplified using the primers 5′-TATTGTTGCTGTGGGACCTG-3′ and 5′-GTTCACATACTTGCCATCGG-3′ at an annealing temperature of 59 °C for 30 cycles. The 202 bp PCR products were digested with MboI (Thermo Fisher Scientific, Waltham, MA, USA) and analyzed by 4% agarose gel electrophoresis. Two bands (160 bp and 42 bp) indicate the *PON1-192Q* allele while three bands (132, 42, and 28 bp) signify the *PON1-192R* allele.

The mouse *Pon1* locus was genotyped by PCR as described [13] using the forward primer p1 (5′-TGGGCTGCAGGTCTCAGGACTGA-3′), exon 1 reverse primer p2 (5′-ATAGGAAGACCGATGGTTCT-3′), and neomycin cassette reverse primer p3 (5′-TCCTCGTGCTTTACGGTATCG-3′) [13]. The 144 bp amplicon from the *Pon1*^+/+^ wild-type allele (obtained with p1, p2 primers) and the 240 bp amplicon from the *Pon1*^−/−^ allele (obtained with p1, p3 primers) were separated on 1.5% agarose gels and visualized by staining with SYBRSafe (Invitrogen).

### 2.5. PON1 Activity Assays

PON1 activity was quantified in human and mouse sera using paraoxon or phenyl acetate as substrates as previously described [6,9,11,24].

### 2.6. Label-Free Mass Spectrometry

Ten micrograms of human or mouse plasma protein were diluted in 50 mM ammonium bicarbonate (15 μL), reduced with dithiothreitol (5.5 mM, 5 min, 95 °C), and alkylated with iodoacetamide (5 mM, 20 min, 25 °C, in the dark). Proteins were digested with Promega sequencing-grade trypsin (0.2 µg, overnight, 37 °C), separated using the Dionex UltiMate 3000 RSLC nanoLC System and analyzed using Q Exactive Orbitrap mass spectrometer (Thermo Fisher Scientific) as previously described [25].

### 2.7. Data Analysis

Datasets were imported into MaxQuant 1.5.3.30 version. Proteins were identified using UniProt human/mouse database with a precision tolerance 10 ppm for peptide masses and 0.08 Da for fragment ion masses as previously described [25]. The MaxQuant data were filtered for reverse identifications (false positives), contaminants, and proteins “only identified by site”. Perseus software (version 1.4.1.3, MPIB, Martinsried, Germany) was used for label-free quantification (LFQ) of peptide intensities. The mean LFQ intensities +/- standard deviation were calculated for each identified differentially expressed protein. The fold change values (FC) for the *PON1* genotype-affected proteins were calculated from the mean LFQ intensities for each genotype.

### 2.8. Statistics

Proteins were considered to be *PON1* genotype-dependent if at least two peptides with >99% confidence were identified and the FC between *PON1* genotype groups was statistically significant (*p* < 0.05). Numeric data were log-transformed and filtered. For multiple comparisons, one-way analysis of variance (ANOVA) with a Bonferroni correction for multiple testing was carried out. *PON1* genotype-affected proteins were normalized using Z-score algorithm. *T*-test was used for comparisons between two groups with *p* values < 0.05 considered significant. Multivariate analyses were carried out by untargeted principal component analysis (PCA) [25].

### 2.9. Bioinformatics Analysis

The datasets containing differentially expressed proteins were uploaded into the IPA Knowledge database. Biological pathways and networks involving *PON1* genotype-affected proteins were identified using the Ingenuity Pathway Analysis resources (IPA, Ingenuity Systems, Mountain View, CA, USA) as previously described [25].

## 3. Results

The effects of the *PON1* genotype on PON1 activity and protein levels in humans and mice are shown in Table 1. As expected, deletion of the *Pon1* gene had a severe effect on Pon1 protein and activity levels in mice, with essentially complete absence of both Pon1 protein and activity [6,13,22]. In contrast, the *PON1-Q192R* polymorphism on PON1 function in humans was relatively mild with about 40% reduction in PON1 protein and fivefold reduction in PON1 activity.

In each group of human participants and mice, label-free nanoLC-MS/MS mass spectrometry identified 196–198 plasma proteins with a minimum of two peptides and 1% false discovery rate (FDR). Proteome Discoverer (PD) analysis showed >90% overlap at the protein level between duplicate runs.

The variation between samples in terms of global proteomic profiles was assessed using the principal component analysis (PCA). There was a clear difference in PCA profiles between *Pon1*^−/−^ mice and their *Pon1*^+/+^ siblings (Figure 1A). However, there was an overlap in PCA profiles between humans with *PON1-192QQ, PON1-192QR,* and *PON1-192RR* genotypes (Figure 1B).

We identified 50 differentiating proteins affected by the *Pon1^−/−^* genotype in mice and 21 differentiating proteins affected by the *PON1-Q192R* polymorphism in humans (Figure 2). Of these, 41 proteins (84%) were affected only in mice, while 12 proteins (57%) were affected only in humans. There were nine proteins, accounting for 43% and 22% of the differentiating proteins in humans and mice, respectively, that were affected by the *PON1* genotype in both species (Figure 2). The differentiating proteins and their functions are listed in Appendix A.

### 3.1. Plasma Proteins Affected by Pon1 Genotype in Mice

Of the 50 proteins differentiating between *Pon1*^−/−^ and *Pon1*^+/+^ mice, 31 (62%) were significantly upregulated (from 1.11-fold for prothrombin F2 to 4.50-fold for haptoglobin HP) and 20 (40%) were downregulated (from 0.02-fold for Pon1 and 0.030-fold for bisphospoglycerate mutase Bpgm to 0.90-fold for inter-α-trypsin inhibitor chain H1, Itih1) (Appendix A). The majority of proteins affected by the mouse *Pon1*^−/−^ genotype are involved in the immune response (*n* = 19, 38%; immunoglobulins Igk, Ighm, Igj, and Igl), lipoprotein metabolism (*n* = 7, 14%; ApoA1, ApoB, ApoC1, ApoD, ApoM, Lcat, Pon1), complement/coagulation cascades (*n* = 8, 16%; Al182371, Cfh, Clu, F2, Fetub, Klkb1, Mbl1, Serpinc1), blood coagulation (*n* = 3, 6%; Hrg, Itih1, F13b), and acute phase response (*n* = 4, 8%; Ambp, Hp, Hpx, Ttr) (Figure 3A, Table 2). Other proteins (*n* = 9; 18%) are involved in glucose/energy metabolism (*n* = 3; Bpgm, Aldoa, Ldha) and other processes (*n* = 6).

### 3.2. Plasma Proteins Affected by PON1-Q192R Polymorphism in Humans

Among the 21 proteins affected by the *PON1-Q192R* polymorphism in humans, five were significantly upregulated (from 1.08-fold for *N*-acetyl-muramoyl-l-alanine amidase PGLYRP2 to 1.32-fold for properdin CFP) and eight were significantly downregulated (from 0.62- for PON1 to 0.95-fold for plasminogen PLG) by the *PON1-192QQ* genotype, compared with *PON1-192QR* and/or *PON1-192QR+PON1-192RR* genotypes (Appendix A). Two proteins (IGHG3 and ITIH3) were significantly upregulated (1.30- and 1.45-fold, respectively) while four (F13B, SERPINA10, RBP4, PON1) were significantly downregulated (0.60- to 0.90-fold) by *PON1-192QQ*, compared with the *PON1-192RR* genotype. Six proteins differentiated between *PON1-192QR+PON1-192RR* genotypes: four were significantly upregulated (APOD, APOM, GPX3, IGL V2-17) and two (SERPINA10, VTN) were downregulated.

Proteins affected by the human *PON1-Q192R* polymorphism are involved in lipoprotein metabolism (*n* = 4, 19%; APOA1, APOD, APOM, PON1), blood coagulation (*n* = 3, 14%; PLG, SERPINA10, F13B) and complement/coagulation cascades (*n* = 4, 19%; C9, V2-17, VTN, FETUB), acute phase response (*n* = 3, 14%; ITIH3, HP, TTR), and immune response (*n* = 5, 24%; immunoglobulins IGL and IGHG3, properdin CFP, *N*-acetyl-muramoyl-l-alanine amidase PGLYRP2, and vitronectin VTN) (Figure 3B, Table 2). Other proteins (*n* = 2, 10%) are involved in redox defense (GPX3) and cardiac muscle development (RBP4).

### 3.3. Overlap between Proteins Affected by PON1 Genotype in Humans and Mice

We identified nine proteins whose levels were affected both by the *PON1-Q192R* polymorphism in humans and the *Pon1*^−/−^ genotype in mice (Figure 2, Table 2). The nine shared proteins represented 43% and 18% of the total number of proteins affected by *PON1* genotype in humans and mice, respectively. Of those common proteins, five were affected in the same direction in humans and mice (i.e., were either upregulated (APOD, APOM) or downregulated (APOA1, F13B, PON1) in both species). Four other proteins were affected in a different direction in humans and mice: FETUB, HP, and TTR were downregulated in humans and upregulated in mice, while IGHG3 was upregulated in humans and downregulated in mice.

### 3.4. Bioinformatics Analysis

To identify biological pathways linked to proteins affected by the *PON1* genotype in humans and mice, bioinformatics analysis with IPA resources was carried out. We found that proteins affected by the human *PON1-Q192R* polymorphism were significantly enriched in 11 canonical pathways, which are linked to atherosclerosis, thrombosis, and Alzheimer’s disease (Figure 4). Proteins in 10 of those pathways were also significantly enriched in *Pon1*^−/−^ mice.

Overall, enrichment ratios were higher in *Pon1*^−/−^ mice than in *PON1-192QQ* humans. The coagulation system, FXR/RXR, LXR/RXR activation, and atherosclerosis signaling pathways had the highest enrichment ratio in *Pon1*^−/−^ mice (0.114, 0.095, 0.091, and 0.064, respectively; −log(*p*-value) = 6.7, 18.6, 16.6, and 10.9, respectively), about threefold higher than in *PON1-192QQ* humans (0.029, 0.040, 0.033, and 0.016, respectively: −log(*p*-value) = 1.9, 9.2, 7.0, and 3.0, respectively) (Figure 4), except for the complement system pathway, which had a similar enrichment ratio in *Pon1*^−/−^ mice and in *PON1-192QQ* humans (0.027; −log(*p*-value) = 1.27 and 1.84, respectively).

One canonical pathway, “Neuroprotective Role of THOP1 in Alzheimer’s Disease”, was marginally significantly enriched in humans (−log (*p*-value) = 1.35; *p* = 0.045) but not in mice (−log (*p*-value) = 0.796; *p* = 0.160) (Figure 4).

Fifteen other pathways contained proteins significantly enriched only in *Pon1*^−/−^ mice and affected glucose/energy metabolism, iron homeostasis, and immune system.

### 3.5. Human PON1-Q192R Polymorphism

IPA identified two top-scoring biological networks associated with human *PON1-Q192R* polymorphism: “Lipid Metabolism, Molecular Transport, Small Molecule Biochemistry” and “Cardiovascular Disease, Neurological Disease, Organismal Injury and Abnormalities” (Table 3).

The “lipid metabolism, molecular transport, small molecule biochemistry” network, identified from analyses *PON1-192QQ* vs. *QR+RR*, had a score of 27 and contained 35 molecules, including 27 proteins. Nine proteins from this network were quantified by label-free mass spectrometry, while 17 proteins and 9 other molecules were identified by IPA to interact with the quantified proteins (Table 3). Graphical illustration of this network is shown in Figure 5A. Proteins in this network participate in lipid metabolism and acute phase/immune response and show strong interactions centering on the lipoproteins LDL and PON1/HDL, the cytokine IL6, and TGFB1. Similar lipid metabolism networks were identified from analyses *PON1- PON1-192QQ* vs. *QR* and *PON1-192QQ* vs. *RR* (Table 3), suggesting that this network was associated with *PON1-192QQ* polymorphism.

The “cardiovascular disease, neurological disease, organismal injury and abnormalities” network, identified from analysis *PON1-192QQ* vs. *RR*, was associated with *PON1-192RR* polymorphism. The cardiovascular/neurological disease network has a score of 14 and contains 35 molecules, including 31 proteins. Five proteins from this network were quantified by label-free mass spectrometry while 26 proteins and 4 other molecules were identified by IPA to interact with the quantified proteins (Table 3). Proteins of this network participate in oxidative stress response, lipoprotein metabolism, and NF-κB signaling and show strong interactions centering on the NF-κB inhibitor alpha NFKBIA and HDL (Figure 5B).

### 3.6. Pon1^−/−^ Mouse Genotype

IPA identified three top-scoring biological networks associated with mouse *Pon1*^−/−^genotype (Table 4). The “Lipid Metabolism, Molecular Transport, Small Molecule Biochemistry” network, with a score of 41, contains 32 proteins. Eighteen of those proteins were quantified by label-free mass spectrometry while 14 were identified by IPA to interact with the quantified proteins. This network involves proteins participating in lipid metabolism, iron metabolism, acute phase response, and complement/coagulation cascades that show strong interactions centering on lipoproteins PON1/APOA1/HDL and APOB/LDL, and ERK1/2 (Figure 6A).

The “Humoral Immune Response, Inflammatory Response, Protein Synthesis” network with a score of 26 contains 36 proteins. Twelve of those proteins were quantified by label-free mass spectrometry while 25 were identified by IPA to interact with the quantified proteins. This network contains immunoglobulins and other proteins participating in the acute phase/immune response, which show strong interactions centering on ERK, P38/MAPK, Akt, and NF-κB (Figure 6B).

The “Cell-to-Cell Signaling and Interaction, Hematological System Development and Function, Immune Cell Trafficking” network with a score of 26 contains 33 proteins. Twelve of those proteins were quantified by label-free mass spectrometry while 21 were identified by IPA to interact with the quantified proteins. This network involves proteins participating in cell signaling and acute phase/immune response, which show strong interactions centering on Tgfb1, Vegf, Jnk (Figure 6C).

## 4. Discussion

The present study provides evidence that genetic reduction of PON1 levels induces proatherogenic changes in plasma proteomes of humans and mice. Specifically, we show for the first time that (i) *PON1-Q192R* polymorphism in humans and *Pon1*^−/−^ genotype in mice induce similar changes in the plasma proteome, which affect a major biological network involving proteins participating in lipoprotein metabolism, the “Lpid Metabolism, Molecular Transport, Small Molecule Biochemistry” network; (ii) these genetic variants also induce other changes in plasma proteomes, which are species-specific and affect different biological networks in humans and mice: “Cardiovascular Disease, Neurological Disease, Organismal Injury and Abnormalities” in *PON1-192QQ* vs. *RR* humans and “Humoral Immune Response, Inflammatory Response, Protein Synthesis” and “Cell-to-Cell Signaling and Interaction, Hematological System Development and Function, Immune Cell Trafficking” in *Pon1*^−/−^ vs. *Pon1*^+/+^ mice. Comparative proteomics of the human *PON1-Q192R* polymorphism and the mouse *Pon1*^−/−^ genotype have not been examined before, and, to our best knowledge, this is the first study of plasma proteomes in *PON1-Q192R* humans and *Pon1*^−/−^ mice. Overall, these changes in plasma proteomes are proatherothrombotic and are known to be associated with human cardiovascular and neurological diseases.

Some of the plasma proteins affected by the *PON1* genotype are shared between humans and mice while changes in other proteins are species-specific and limited to humans or mice. The shared proteins include lipoproteins APOD and APOM (negative regulators of lipoprotein oxidation), the carrier of hydrophobic molecules AFM (involved in transport of fatty acids, or vitamin E when the lipoprotein system is insufficient), the negative regulator of endopeptidase activity FETUB, the antioxidant HP (involved in acute inflammatory response), the thyroid hormone-binding protein TTR (involved in thyroxine transport from the blood to the brain), the fibrin-stabilizing factor F13B (associated with a risk of stroke), and the immunoglobulin IGHG3 (involves B-cell signaling, complement activation, and humoral immunity). The species-specific proteins affected by the *PON1* genotype are more numerous in mice (*n* = 41) than in humans (*n* = 12) and most likely reflect a much more severe effect of the *Pon1* gene deletion on Pon1 function (essentially complete absence of both Pon1 protein and activity) in mice [6,13,22] than the effect of *PON1-Q192R* polymorphism on PON1 function in humans (about 40% reduction in PON1 protein and 10-fold in PON1 activity).

Our findings in *PON1-Q192R* humans and *Pon1*^−/−^ mice can explain the cardiovascular and neurological pathologies associated with the variation in *PON1* genotypes in the two species. Specifically, identification of *PON1* genotype-responsive proteins involved in lipid/cholesterol metabolism/transport (APOD, APOM in humans; ApoA1, ApoB, ApoC1, ApoD, ApoM in mice), acute phase response (HP, ITIH3, TTR in humans; Ambp, Hpx, Hp, Ttr in mice), and complement/coagulation (C9, PLG, SERPINA10 in humans; Cfh, F2, Klkb1, Mbl1, Serpinc1, Itih1 in mice) provides support for this conclusion.

We found that the lipoproteins APOD and APOM were upregulated by *PON1-192QR* vs. *PON1-192RR* genotype in humans and *Pon1*^−/−^ genotype in mice. Elevation in APOD expression has been associated with a number of pathological conditions including neurodegenerative disease [26]. Both APOD and APOM are negative regulators of lipoprotein oxidation and have antiatherogenic properties. ApoD is an acute response protein with a protective function mediated by the control of peroxidized lipids. In mice, deletion of the *ApoD* gene increases the sensitivity to oxidative stress, levels of lipid peroxides in the brain, and impairs learning and locomotor abilities [26]. APOM binds oxidized phospholipids in plasma and increases the antioxidant effect of HDL [27]. Thus, changes in APOD and APOM levels could contribute to the association of the *PON1* genotype with cardiovascular and neurological diseases.

In the present study we found that ApoA1 and ApoC1, known to be negative regulators of lipid/cholesterol catabolism/transport cholesterol, were downregulated, while ApoB, a positive regulator of lipid/cholesterol storage, was upregulated in *Pon1*^−/−^ mice. These findings suggest that the *Pon1*^−/−^ genotype exerts a proatherogenic effect on the lipoprotein homeostasis.

We identified several proteins of the coagulation system that were affected by the *PON1* genotype. For example, PLG, SERPINA10, and F13B were downregulated by the *PON1-192QQ* genotype in humans. PLG, bound to fibrin clots, is converted to plasmin, which plays a crucial role in dissolving blood clots. SERPINA1, a negative regulator of blood coagulation, inhibits the activated coagulation factors X and XI, thus preventing fibrin clot formation [28]. F13B is a subunit of prototransglutaminase, which is activated at the final stage of coagulation and which stabilizes the clot by forming an amide bond between Glu and Lys residues of fibrin [29]. A genetic variant of F13B is associated with the risk of stroke [30]. Thus, reduced levels of PLG, SERPINA10, and F13B in carriers of *PON1-192QQ* genotype suggest that the 192QQ variant could increase the propensity to atherothrombosis, which might account for an association of this variant with vascular disease [16,31].

The antithrombin Serpinc1, a component of the complement/coagulation cascades, was downregulated in *Pon1*^−/−^ mice. As genetic variants of SERPINC1 are known to be associated with venous thrombosis in humans [32], the downregulation of the antithrombin Serpinc1 in *Pon1*^−/−^ mice could increase thrombin activity, thereby increasing blood clotting, which might account for increased atherosclerotic lesions observed in these animals when fed with a high-fat diet [13].

Proteins affected by *PON1-Q192R* polymorphism in humans and *Pon1*^−/−^ genotype in mice were also significantly enriched in the LXR/RXR activation pathway, which can also affect blood homeostasis (Figure 4). RXR is a negative regulator of platelet function/aggregation and thrombus formation [33]. Thus, proteins of the LXR/RXR pathway that we found to be affected by the *PON1-Q192R* polymorphism in humans (C9, FETUB, HP, PON1, RBP4, TTR, APOM) and *Pon1*^−/−^ genotype in mice (Alb, Ambp, ApoA1, ApoB, ApoD, ApoM, Clu, FetuB, Hpx, Lcat, Pon1, Ttr) can contribute to the cardiovascular phenotype associated with the *PON1* genotype.

We also found that the *PON1* genotype affected proteins of the Il-7 signaling pathway. IL-7 activates JAK-STAT, PI-3 kinase, and Src kinase pathways, which is important in the development/proliferation of the immune B and T cells in mice and T cells in humans. *Pon1*^−/−^ mice showed significantly upregulated expression of immunoglobulins such as Igha, Ighm, Ighg (1, 3), Ighv (1-76, 3-6, 7-1, 10-1), and Igkv (4-63, 8-28, 17-127, 19-93) (Table 2, Appendix A). *PON1-192QQ* humans also showed upregulated expression, although to a more limited extent than *Pon1*^−/−^ mice, with only two immunoglobulins upregulated, IGHG3 and IGLV2-17. These findings suggest genetic attenuation of PON1 activity leads to enhanced immune response due to increased Il-7/B-cell signaling.

## 5. Conclusions

Our findings identify a proatherogenic phenotype in the plasma proteome associated with the *PON1* genotype in humans and mice. This phenotype includes mild dysregulation of lipoproteins but is silent, even in *Pon1*^−/−^ mice. However, it can be exacerbated by a stress such as severe dyslipidemia induced by a proatherogenic high-fat diet. Indeed, *Pon1*^−/−^ mice became susceptible to aortic lesions only when fed with a high-fat diet but not a standard chow diet [13]. Other metabolic stressors, such as elevated Hcy, have been found to modulate expression of brain proteins involved in oxidative stress and neurodegeneration in *Pon1*^−/−^ mice [34].

## Figures and Tables

**Figure 1 antioxidants-09-01198-f001:**
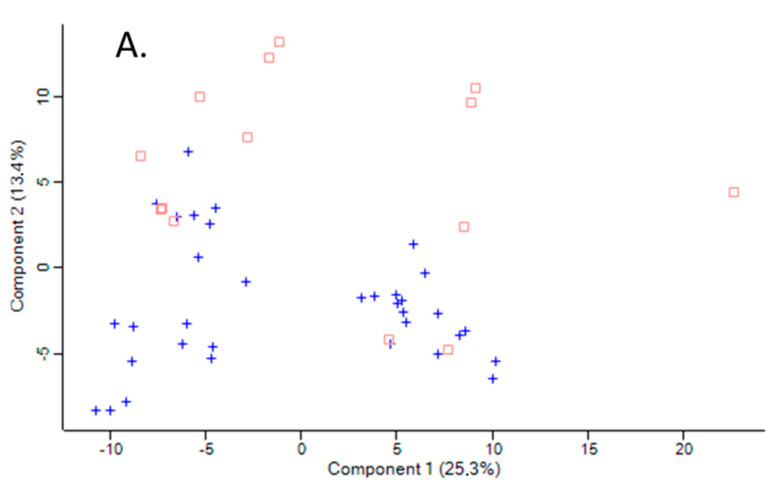
Principal component analysis of the LFQ intensities for plasma proteins. (**A**). Mice: *Pon1*^−/−^ (*n* = 17; blue cross) and *Pon1*^+/+^ siblings (*n* = 8; red square). (**B**). Humans: *PON1-192QQ* (*n* = 51; blue cross), *PON1-192QR* (*n* = 30; green circle), and *PON1-192RR* humans (*n* = 19; red square). Calculations were performed with Perseus.

**Figure 2 antioxidants-09-01198-f002:**
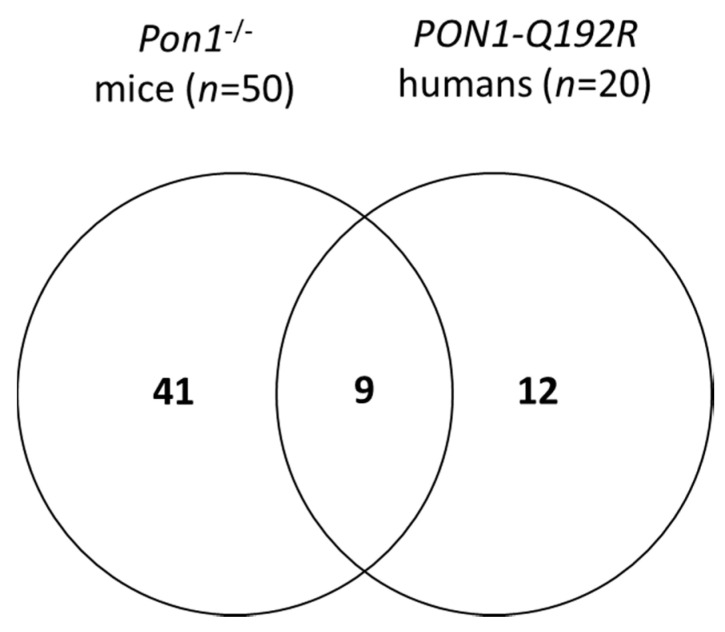
Venn diagram illustrating a partial overlap between proteins affected by PON1 genotype in humans and mice.

**Figure 3 antioxidants-09-01198-f003:**
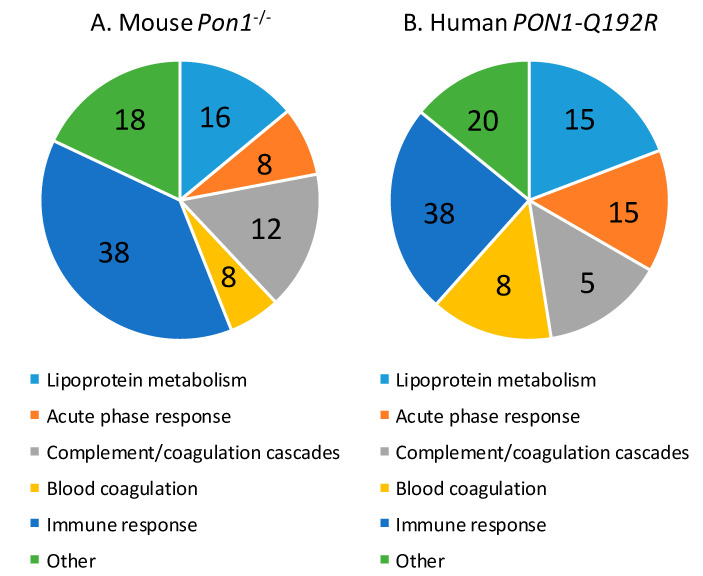
Relative numbers of proteins (%) involved in the indicated molecular processes affected in *Pon1*^−/−^ mice (**A**) and *PON1*-*Q192R* humans (**B**).

**Figure 4 antioxidants-09-01198-f004:**
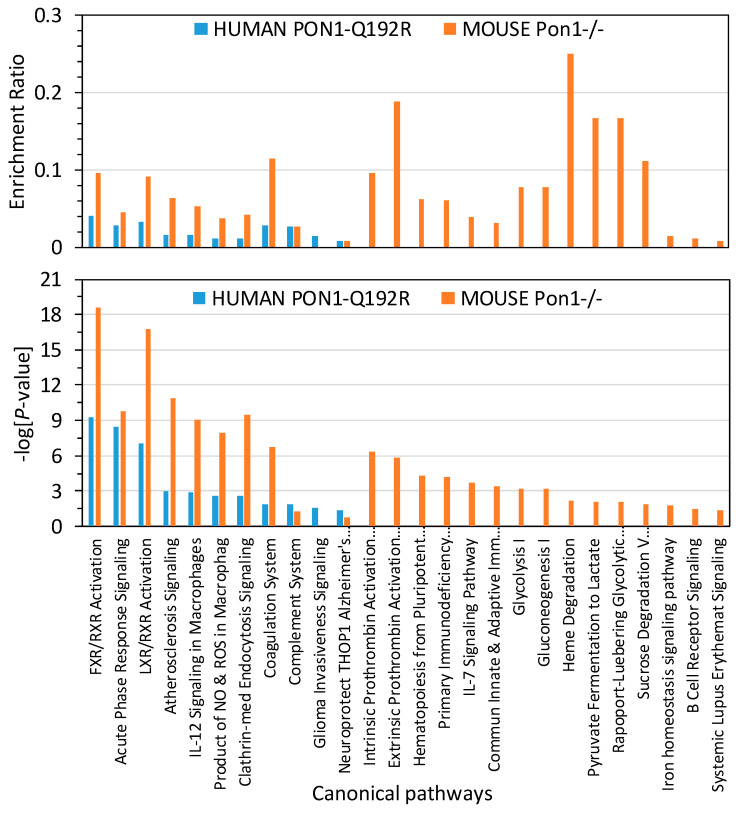
Enrichment ratios and canonical pathways of differentially expressed proteins in *PON1-192QQ* vs. *PON1-192RR+QR* humans and *Pon1*^−/−^ vs. *Pon1*^+/+^ mice identified by IPA. Benjamini–Hochberg, Benferroni, and false discovery rate corrections were applied to minimize the number of false positives.

**Figure 5 antioxidants-09-01198-f005:**
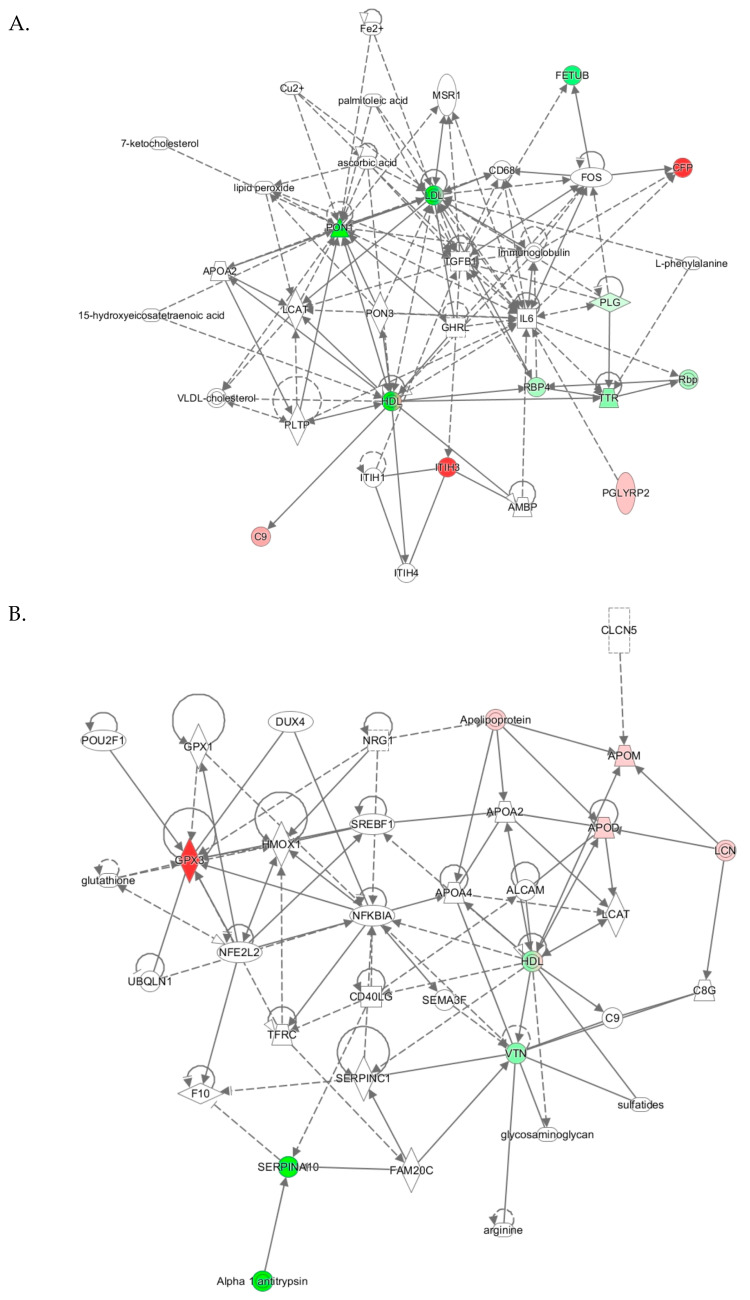
Molecular networks associated with *PON1-Q192R* polymorphism in humans. (**A**) Lipid Metabolism, Molecular Transport, Small Molecule Biochemistry; (**B**) Cardiovascular Disease, Neurological Disease, Organismal Injury and Abnormalities. Upregulated and downregulated proteins are highlighted in red and green, respectively.

**Figure 6 antioxidants-09-01198-f006:**
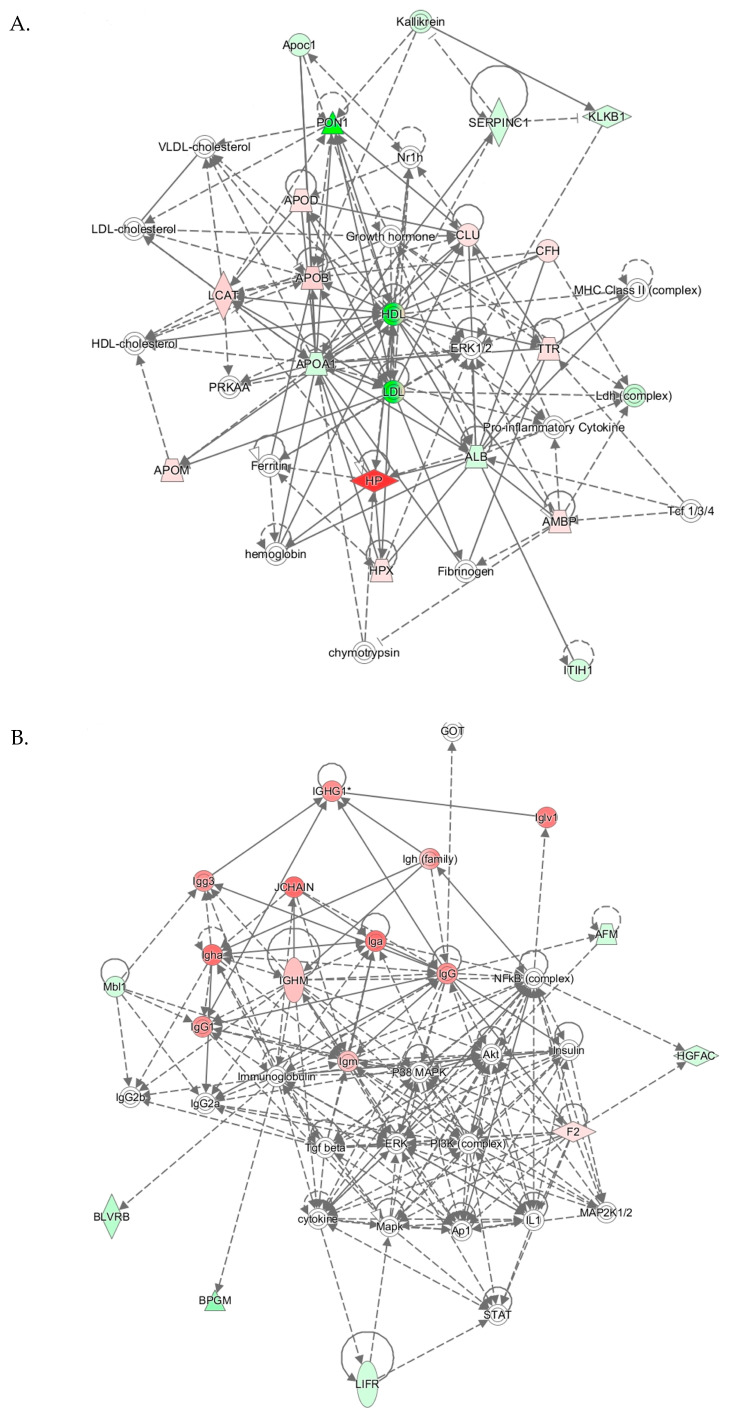
Molecular networks associated with *Pon1*^−/−^ genotype in mice. (**A**) Lipid Metabolism, Molecular Transport, Small Molecule Biochemistry. (**B**) Humoral Immune Response, Inflammatory Response, Protein Synthesis. (**C**) Cell-to-Cell Signaling and Interaction, Hematological System Development and Function, Immune Cell Trafficking. Upregulated and downregulated proteins are highlighted in red and green, respectively.

**Table 1 antioxidants-09-01198-t001:** *PON1* genotype, activity, and protein levels in humans and mice.

Human Paraoxonase 1			Mouse Paraoxonase 1		
Genotype (n)	Activity ^a^	Protein ^b^	Genotype (*n*)	Activity ^a^	Protein ^b^
*PON1-192RR* (19)	100	100	*Pon1*^+/+^ (17)	100	100
*PON1-192QR* (30)	20.6	63.0			
*PON1-192QQ* (51)	14.3	60.0	*Pon1*^−/−^ (8)	0.0	2.0

^a^ Relative mean values determined in serum using paraoxon as a substrate. ^b^ Relative mean values calculated from label-free mass spectrometry data for PON1 shown in Appendix A.

**Table 2 antioxidants-09-01198-t002:** Proteins affected by *PON1* genotype in mice and humans *.

Unique to Mice (*n* = 41) ^$^			Unique to Humans (*n* = 12) ^$^	Proteins Affected in Mice and Humans (*n* = 9) ^#^
↓Afm	↓Blvrb	↓Ldha	↑GPX3	
↓Alb	↓Bpgm	↓Lifr	↓RBP4	
↓Aldoa	↓Ica	↑Mug1		
Lipoprotein metabolism (*n* = 3): *↑*ApoB, *↓***ApoC1**, *↑*Lcat				Lipoprotein metabolism (*n* = 3): ↓APOA1↓, *↑APOD↑*, *↑***APOM***↑*, *↓PON1*↓
Acute phase response (*n* = 2): ↑Ambp, ↑Hpx			Acute phase response (*n* = 1): ↑ITIH3,	Acute phase response (*n* = 2): ↑HP↓, *↑***TTR**↓
Blood coagulation (*n* = 2): ↑Hrg, ↓Itih1			Blood coagulation (*n* = 3): ↓PLG, ↓SERPINA10, ↓VTN	Blood coagulation (*n* = 1): ↓F13B↓
Complement/coagulation (*n* = 7): ↑Al182371, ↑Cfh, ↑Clu, ↑**F2**, ↓Klkb1, ↓Mbl1; ↓Serpinc1			Complement/coagulation (*n* = 2): ↑C9, ↑V2-17 (IGL)	Complement/coagulation (*n* = 1): ↑FETUB↓
Immune response (n =18): ↑Igh (*n* = 9), ↑Igj, ↑Igk (*n* = 6), ↑Igl (*n* = 2)			Immune response (*n* = 5): ↑CFP, ↓N/A, ↑PGLYRP2, ↑V2-6 (IGL),	Immune response (*n* = 1): ↓IGHG3↑

* Proteins highlighted in bold are associated with stroke subtypes in humans (Sikora et al., 2019). **^$^** Up and down arrows indicate the direction of change in protein levels. ^#^Arrows left and right to the protein acronym refer to the change in protein levels in mice and humans, respectively.

**Table 3 antioxidants-09-01198-t003:** Top molecular networks associated with human *PON1-Q192R* polymorphism. Upregulated (**↑**) and downregulated (**↓**) proteins are highlighted in bold red and green, respectively. Graphical illustrations of interactions between proteins in these networks are shown in Figure 5.

Analysis	Molecules in Network	Score	Focus Molecules	Top Diseases and Functions
*PON1-192QQ* vs. *QR+RR* (Figure 5A)	15-hydroxyeicosatetraenoic acid, 7-ketocholesterol, AMBP, APOA2, ascorbic acid, ↑**C9**, CD68, ↑**CFP**, Cu2, Fe2, ↓**FETUB**, FOS, GHRL, HDL, IL6, Immunoglobulin, ITIH1, ↑**ITIH3**, ITIH4, L-phenylalanine, LCAT, LDL, lipid peroxide, MSR1, palmitoleic acid, ↑**PGLYRP2**, ↓**PLG**, PLTP, ↓**PON1**, PON3, Rbp, ↓**RBP4**, TGFB1, ↓**TTR**, VLDL-cholesterol	27	9	Lipid Metabolism, Molecular Transport, Small Molecule Biochemistry
*PON1-192QQ* vs. *QR*	15-hydroxyeicosatetraenoic acid, APCS, ↓**APOA1**, Apoc1, APOF, APOL1, ↓**APOM**, bilirubin, ↑**C9**, ↑**CFP**, Cxcl9, Fe2, Ferritin, ↓**GPX3**, Growth hormone, GSTT1, ↓**HBB**, HBD, HBG1, HBQ1, HDL, hemoglobin, ↓**HPR**, Immunoglobulin, Insulin, ITIH4, LDL, ↓**PON1**, PON3, Rbp, ↓**RBP4**, SAA2, SELENOT, ↓**TTR**, IGHV1-69, ↑**IGLV3-9**	28	11	Lipid Metabolism, Molecular Transport, Small Molecule Biochemistry
*PON1-192QQ* vs. *RR*	15-hydroxyeicosatetraenoic acid, AFM, Alpha 1 antitripsin, AMBP, C1QTNF3, CCR2, CD40LG, Cd64, CD68, CXADR, ↓**F13B**, FCGR2C, Fe2, IFNG, IgG3 kappa, IgG3 lambda, IGHG1, ↑**IGHG3**, IGHG4, ITIH1, ↑**ITIH3**, ITIH4, LCAT, lipid peroxide, MSR1, MTRR, palmitoleic acid, ↓**PON1**, PON3, RAD51AP1, ↓**RBP4**, ↓**SERPINA10**, TGFB1, TNFAIP6, VLDL-cholesterol	18	6	Lipid Metabolism, Molecular Transport, Small Molecule Biochemistry
*PON1-192QR* vs. *RR* (Figure 5B)	ALCAM, Alpha 1 antitrypsin, APOA2, APOA4, ↑**APOD**, Apolipoprotein, ↑**APOM**, arginine, C8G, C9, CD40LG, CLCN5, DUX4, F10, FAM20C, glutathione, glycosaminoglycan, GPX1, ↑**GPX3**, HDL, HMOX1, LCAT, LCN, NFE2L2, NFKBIA, NRG1, POU2F1, SEMA3F, ↓**SERPINA10**, SERPINC1, SREBF1, sulfatides, TFRC, UBQLN1, ↓**VTN**	14	5	Cardiovascular Disease, Neurological Disease, Organismal Injury and Abnormalities

**Table 4 antioxidants-09-01198-t004:** Top molecular networks associated with *Pon1*^−/−^genotype in mice. Upregulated (**↑**) and downregulated (**↓**) proteins are highlighted in bold red and green, respectively. Graphical illustrations of interactions between proteins in these networks are shown in Figure 6A–C.

Analysis	Molecules in Network	Score	Focus Molecules	Top Diseases and Functions
*Pon1*^−/−^ vs. *Pon1*^+/+^Figure 6A	↓**ALB**, ↑**AMBP**, ↓**APOA1**, ↑**APOB**, ↓**Apoc1**, ↑**APOD**, ↑**APOM**, ↑**CFH**, chymotrypsin, ↑**CLU**, ERK1/2, Ferritin, Fibrinogen, Growth hormone, HDL, HDL-cholesterol, hemoglobin, ↑**HP**, ↑**HPX**, ↓**ITIH1**, Kallikrein, ↓**KLKB1**, ↑**LCAT**, ↓**Ldh** (complex), LDL, LDL-cholesterol, MHC Class II (complex), Nr1h, ↓**PON1**, PRKAA, Pro-inflammatory Cytokine, ↓**SERPINC1**, Tcf 1/3/4, ↑**TTR**, VLDL-cholesterol	41	17	Lipid Metabolism, Molecular transport, Small Molecule Biochemistry
*Pon1*^−/−^ vs. *Pon1*^+/+^Figure 6B	↓**AFM**, Akt, Ap1, ↓**BLVRB**, ↓**BPGM**, cytokine, ERK, F2, GOT, ↓**HGFAC**, Iga, IgG, IgG1, IgG2a, IgG2b, Igg3, Igh (family), ↑**Igha**, IGHG1, ↑**IGHM**, ↑Iglv1, Igm, IL1, Immunoglobulin, Insulin, JCHAIN, ↓**LIFR**, MAP2K1/2, Mapk, ↓**Mbl1**, NFkB (complex), P38 MAPK, PI3K (complex), STAT, Tgf beta	26	12	Humoral Immune Response, Inflammatory Response, Protein Synthesis
*Pon1*^−/−^ vs. *Pon1*^+/+^Figure 6C	↓**ALDOA**, ANGPT2, CASR, CD163, EED, ↓**F13B**, ↑**FETUB**, FN1, ↑**HP**, ↑**Hrg**, ↑**Igha**, ↓**Ighg3**, ↑**Ighv3-6**, Igkv1-117, Igkv14-111, ↑**Igkv17-127**, IL4, Jnk, LDH (family), ↓**LDHA**, LINC01139, lipid peroxide, lysophosphatidylinositol, miR-18a-5p (and other miRNAs w/seed AAGGUGC), MSR1, Mug1/Mug2, Pkc(s), PKD1, PLAGL2, pyruvaldehyde, SBNO2, TGFB1, TLL1, trypsin, Vegf	26	12	Cell-to-Cell Signaling and Interaction, He- matological System Development and Function, Immune Cell Trafficking

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
