# Peer review of "Genetic Attenuation of Paraoxonase 1 Activity Induces Proatherogenic Changes in Plasma Proteomes of Mice and Humans"

_antioxidants, 2020, doi:10.3390/antiox9121198_

Round 1
Reviewer 1 Report
This is a very interesting and well written study that provides new information relevant to understanding the molecular basis of the relationship between PON1 and arteriosclerosis. I recommend publication as is.
Author Response
We thank Reviewer 1 for positive comments on our manuscript.
Reviewer 2 Report
The manuscript by Sikora and colleagues describes the effect of PON1 genotype on the plasma proteome of mice and humans. A polymorphism in human PON1 and deletion of PON1 in mice induced similar changes in the plasma proteome. The authors conclude that PON1 interacts with molecular pathways that influence lipoprotein metabolism, inflammation and coagulation. The observed changes in biological networks are known to be associated with increased risk for cardiovascular and neurological diseases.
There are a number of problems with the figures.
Figure 1A and B – there are more data points on the graph than are indicated in the legend.
Figure 1B – Data for PON1-192QR is absent.
Figure 3 – data is presented for CBS-/- mice and patients???
Section 3.3 and Table 2 – TTR does not appear to be downregulated in both species.
Author Response
Comment: Figure 1A and B – there are more data points on the graph than are indicated in the legend.
Response: There are twice as many data points on the graph because principal component analysis generates TWO points for each individual sample.
Comment: Figure 1B – Data for PON1-192QR is absent.
Response: As stated in the legend to Figure 1B, data points for PON1-192QR are indicated by green circles.
Comment: Figure 3 – data is presented for CBS-/- mice and patients???
Response: We thank the Reviewer for pointing out this mistake in the legend to Figure 3. Corrected legend to Figure 3 now states that data for Pon1-/- mice (A.) and for PON1-Q192R humans (B.) are presented.
Comment: Section 3.3 and Table 2 – TTR does not appear to be downregulated in both species.
Response: We thank the Reviewer for pointing out this mistake. The text in section 3.3. has now been corrected to indicate that TTR is upregulated by the Pon1-/- genotype in mice, but downregulated by the PON1-Q192R polymorphism in humans, consistent with data shown in Table 2 and Supplementary Table S1.
Reviewer 3 Report
In the present MS, Authors Sikora et collaborators, investigated impact of genetic attenuation of paraoxonase 1 (PON 1) activity in plasmatic changes in proteomes of mice and human.
By using valuable technics, Authors showed that reduction in PON1 activity in human (PON1 Q192R polymorphism) and mice with Pon1 -/- displayed similar changes in plasma proteomes.
This is an interesting and readable MS. That said I have few points to address:
- As mentioned by the Authors, PON1 was reported to account for HDL antioxidant activity.
Then and as far as Antioxidant journal is concerned, I did not find any specific data on the field of oxidative stress, damage or oxidative biomarker in the MS. Did Authors evaluated antioxidant capacity in the sera of mice/humans?
- In table 1, I am not sure to understand what “Protein” correspond to. Is it total plasmatic protein? PON1 protein quantification? I saw results in table 1 are expressed in percent of the control… Was PON1 activity expressed per protein? Per PON1 specific protein?
- Do Authors have data on mouse with heterozygote genotype for PON1: Pon1 +/-?
- On figure 2, why Authors did not compare Pon1 -/- mice with PON1 Q192Q humans?
Author Response
Comment: Did Authors evaluated antioxidant capacity in the sera of mice/humans?
Response: No, we did not evaluate antioxidant capacity of serum because this aspect of PON1 has already been examined by Bhattacharyya T et al. JAMA 2008;299:1265-1276 (ref 16 in our manuscript). However, we identified one oxidative stress response protein, GPX3, which was affected by PON1 genotype in humans but not in mice.
Comment: In table 1, I am not sure to understand what “Protein” correspond to. Is it total plasmatic protein? PON1 protein quantification? I saw results in table 1 are expressed in percent of the control… Was PON1 activity expressed per protein? Per PON1 specific protein?
Response: PON1 protein quantified by label-free mass spectrometry. This is stepet in footnote b in Table 1.
Comment: Do Authors have data on mouse with heterozygote genotype for PON1: Pon1 +/-?
Response: We did not examine Pon1+/- heterozygotes because of limitations as to the number of samples that can be processed in one run on a mass spectrometry equipment.
Comment: On figure 2, why Authors did not compare Pon1 -/- mice with PON1 Q192Q humans?
Response: Analyses for mice and humans have been carried out in separate experiments. Please note that we can compare only groups that have been examined in the same run in a label-free mass spectrometry experiment. We did not compare Pon1 -/- mice with PON1 Q192Q humans because such experiment would require a separate experiment with much larger number of groups and samples, which would not be logistically feasible due to a limited access to the mass spectrometry equipment, not mentioning the cost.
Reviewer 4 Report
The manuscript by Sikora et al. is a very interesting attempt at identifying differences in the human and mouse plasma proteome caused by a particular paraoxonase-1 polymorphism or lack of PON1, respectively. The authors found novel results that support some of the described relationships between PON1 and disease, importantly, they identified certain target proteins that can lead into further research centered on those proteins in relation to PON1. I find the manuscript of high relevance to the field. One main concern that I have raised below is the use of EDTA to collect plasma (I assume it was mouse plasma, see comment), which irreversibly inhibits PON1 activity. The authors reported measurable PON1 activity in mice, and I am not sure how that was possible. Another minor comment is that there are many important details missing, showing lack of attention to detail when preparing the manuscript, in addition to a few typos identified.
Specific observations
Materials and methods section:
- How was blood collected from the participants? Please, include those details.
- Line 93: Blood was collected from the cheek vein (I assume that is mouse blood, but please, indicate that in the revision) into EDTA-containing tubes. EDTA is a calcium chelator, and therefore, it irreversibly inhibits PON1 activity. I am not sure how the researchers were able to obtain activity values from mouse plasma, as shown in Table 1. These measurements and blood collection should be repeated if indeed EDTA was used.
- Genotyping: The authors should briefly explain how they extracted human DNA.
- PON1 activity section: the authors should specify that they used plasma (or serum) and if the samples used were from mice, human or both.
- Label-free mass spectrometry: There is no explanation regarding which samples were used (mouse and/or human plasma?) and how the samples were prepared for the mass spectrometry analyses (denaturing, DTT, IAA, etc).
Results section:
- Table 1 is missing the units for PON1 activity and protein.
- The legend in Figure 3 needs your attention: “Figure 3. Relative numbers of proteins (%) involved in the indicated molecular processes affected in Cbs-/- mice (A.) and CBS-/- patients (B.).” These mouse strains are not described as being used in the materials and methods section.
- Legend of Figure 5 should specify what A and B are.
Discussion section:
- Line 341: “The species-specific proteins affected in Pon1-/- mice are more numerous in mice (n = 42) than in humans (n = 12)”. I am not sure this sentence makes sense. Did the authors mean to say “the species-specific proteins affected by PON1 genotype are more numerous…”? Also, based on Figure 2, the authors found 41, not 42, proteins specific to mice.
- Lack of discussion of the study limitations.
Supplementary material section:
- Supplementary Table 2 is missing a legend. Please, indicate what the different colors mean.
Author Response
Materials and methods section:
Comment: How was blood collected from the participants? Please, include those details. Line 93: Blood was collected from the cheek vein (I assume that is mouse blood, but please, indicate that in the revision) into EDTA-containing tubes. EDTA is a calcium chelator, and therefore, it irreversibly inhibits PON1 activity. I am not sure how the researchers were able to obtain activity values from mouse plasma, as shown in Table 1. These measurements and blood collection should be repeated if indeed EDTA was used.
Response: We collected blood for into EDTA-tube to obtain plasma, which was used for label-free mass spectrometry experiments. We also collected blood into empty tubes, allowed it to clot and separated serum, which was used for PON1 activity assays.
Comment: Genotyping: The authors should briefly explain how they extracted human DNA.
Response: The genotyping section 2.4. was modified as requested by adding info on DNA extraction.
Comment: PON1 activity section: the authors should specify that they used plasma (or serum) and if the samples used were from mice, human or both.
Response: In section 2.5. we have added info stating that PON1 activity was quantified in human and mouse sera.
Comment: Label-free mass spectrometry: There is no explanation regarding which samples were used (mouse and/or human plasma?) and how the samples were prepared for the mass spectrometry analyses (denaturing, DTT, IAA, etc).
Response: We have used plasma for label-free mass spectrometry. This info as well as details on sample preparation are now included in section 2.6.
Results section:
Comment: Table 1 is missing the units for PON1 activity and protein.
Response: An explanation that relative values for PON1 activity and protein are shown in Table 1 has now bee included.
Comment: The legend in Figure 3 needs your attention: “Figure 3. Relative numbers of proteins (%) involved in the indicated molecular processes affected in Cbs-/- mice (A.) and CBS-/- patients (B.).” These mouse strains are not described as being used in the materials and methods section.
Response: We thank the reviewer for pointing out this mistake on our part, which is now corrected to indicate that Cbs-/- mice and PON1 Q192R humans were analyzed.
Comment: Legend of Figure 5 should specify what A and B are.
Response: We have included the missing description for panels A and B in Figure 5.
Discussion section:
Comment: Line 341: “The species-specific proteins affected in Pon1-/- mice are more numerous in mice (n = 42) than in humans (n = 12)”. I am not sure this sentence makes sense. Did the authors mean to say “the species-specific proteins affected by PON1 genotype are more numerous…”? Also, based on Figure 2, the authors found 41, not 42, proteins specific to mice
Response: We thank the Reviewer for catching these mistakes, which are now corrected.
Changes in the manuscript are highlighted in yellow.
Round 2
Reviewer 3 Report
Authors addressed correctly points I raised.